

# Children are not like other loads: a cross-cultural perspective on the influence of burdens and companionship on human walking

Leah Bouterse[1] and Cara Wall-Scheffler[1,2]

[1] Department of Biology, Seattle Pacific University, Seattle, WA, United States of America
[2] Department of Anthropology, University of Washington, Seattle, WA, United States of America

## ABSTRACT

A major portion of humans' activity-based energy expenditure is taken up by locomotion, particularly walking. Walking behaviors have energetic outcomes and as such can be important windows into how populations and groups adjust to different environmental and task constraints. While sex differences in the speed of paired walkers have been established by others, the dynamics of how walkers adjust their speed in more varied groups and in groups containing children remains unexplored. Furthermore, little ecological data exists to illustrate the relationships between walking speed and child-carrying. Here, we aim to determine how culture impacts the effects of group composition and infant-carrying on walking speed. Because the determinants of group dynamics and parental investment are partially cultural, we examine walking behavior in the Northwestern United States and in Central Uganda. Using an observational method, we recorded the speed, load carriage, and group composition of pedestrians in a single naturalistic urban environment within each country. Our data suggest that children are treated fundamentally differently than other loads or the presence of walking partners, and that major speed adjustments are child-dependent. Our data furthermore indicate that Ugandans walk more slowly in groups than when alone, while Americans walk more quickly in groups. Clear distinctions between the groups make large generalizations about walking behavior difficult, and highlight the importance of culturally specific contexts.

## INTRODUCTION

Today, foraging populations who travel far distances face consistent pressures to decrease the energetic cost of their journeys and even people living in Western contexts clearly make individual speed and gait choices that minimize energy expenditure (*Bertram & Ruina, 2001*; *Ralston, 1958*; *Selinger et al., 2015*; *Srinivasan & Ruina, 2006*; *Zarrugh, Todd & Ralston, 1974*). Theoretically, energy-minimizing behavior is believed to stem from the overarching principle that time and energy spent on one task (e.g., walking) takes away from energy available for reproduction and in the case of social organisms, investment in social groups (*Torrence, 1989*; *Pollard & Blumstein, 2008*; *Wall-Scheffler & Myers, 2013*).

Corresponding author
Cara Wall-Scheffler,
cwallsch@spu.edu,
cwallsch@u.washington.edu

Optimization of locomotor costs then, significantly impacts feeding, foraging, avoiding predators, and reproducing, as well as sociality (*Wagnild & Wall-Scheffler, 2013*). The cost of locomotion has been shown to depend on morphological as well as behavioral conditions. For example, increases in body size and limb length are associated with decreased locomotor cost relative to mass (*Pontzer, 2007*; *Steudel, 1994*). Additionally, males (i.e., bigger individuals with longer lower limbs) walk at faster speeds than smaller females across different populations (*Costa, 2010*; *Wall-Scheffler, 2012*). Behaviors, such as load carrying and walking in groups, also impact the costs and decisions associated with walking and can potentially offer additional selection pressures to single-person free walking (*Wagnild & Wall-Scheffler, 2013*; *Boles, 1981*; *Bastien et al., 2005*; *Wall-Scheffler, Wagnild & Wagler, 2015*).

Like many mammals, human walking shows a speed at which energy expenditure per distance travelled is lower than all other speeds—as demonstrated by the curvilinear relationship between cost of transport (metabolic cost for a given distance) and speed (*Bertram, 2005*; *DiPrampero, 1986*). Thus, when walking alone, humans are expected to prefer speeds that minimize costs under particular conditions; however, when walking in groups, humans do not necessarily walk at the speed that minimizes energy. *Boles (1981)* and *Costa (2010)* show speed changes related to group size and composition in an ecological setting. Dyads walk more slowly than individuals, since one member must adjust for the slower speed of their partner (*Boles, 1981*). *Wagnild & Wall-Scheffler (2013)* suggest that these adjustments are based on gender and the emotional closeness between walkers. While men and women walking together each adjust their speed away from each's individualized energetic optimum, the same does not hold true for individuals that share a romantic relationship. In heterosexual romantic partners, men will walk more slowly than his energetic optima in order to match the walking speed of his partner. When men walk with other men, their speeds tend to increase (*Wagnild & Wall-Scheffler, 2013*; *Costa, 2010*). Social phenomena in some cultures—including a greater degree of social hierarchy in American male groups and American male aversion to showing same-sex intimacy–have been thought to explain this observation (*Costa, 2010*). The relationships between sex and group walking speed hold true for triads as well as dyads, but the effects have been shown to decrease for groups greater than three individuals (*Costa, 2010*).

Load-bearing is another behavior which impacts walking costs and speed. The net metabolic power of walking has been shown to depend on load mass: data show that a unit of load mass is more expensive to transport than a unit of body mass placed at the same position relative to the carrier's center of mass (*Bastien et al., 2005*; *Kramer, 2004*). The energetically optimal walking speed for an individual tends to decrease with increasing load mass in multiple studies (*Wall-Scheffler & Myers, 2013*; *Bastien et al., 2005*). Groups of East African women, though, have been shown to carry loads up to 20% of their body mass without increased costs, potentially due to some change in kinematics (*Maloiy et al., 1986*). Effectiveness of carrying a load seems to depend on its position. Generally, loads are shown to add less metabolic cost the closer they are carried to the center of mass, and more so if they are carried on the back than on the front (*Abe, Yanagawa & Niihata, 2004*; *Watson et al., 2008*, although see *Wall-Scheffler & Myers, 2013* for the low cost of

carrying on the front and *Maloiy et al., 1986*; *Panter-Brick, 1992* for low costs of carrying on the head).

The carriage of children is a special case of load bearing. While regular carriage of heavy food or goods applies primarily to human primates, child loads are common across species. Particularly for mammalian species, care and transport of offspring during lactation presents enormous energetic costs for one parent, often the mother (*Kramer, 2004*; *Ross, 2001*). Despite the metabolic costs of carrying infants, it remains a widespread strategy across mammals (*Rhinegold & Keen, 1963*). Amongst primates, infants either "ride" while grasping to parents' fur or are transported orally (*Ross, 2001*). *Ross (2001)* argues that fur riding has evolved independently and been conserved more than five times in primates. While maternal carrying remains prevalent across primate taxa, many New World Monkey fathers, as well as dominant male baboons, also carry infants (*Rhinegold & Keen, 1963*).

Non-human primate infants are typically carried on the mother's back or front, often relying on the infant's grip. In humans, however, carriage relies on the parents without help from the child's grip (*Rhinegold & Keen, 1963*). Though ecological data on human infant-carrying is limited, Rhinegold and Keen (*Rhinegold & Keen, 1963*) found that, in an American urban center, women carried infants more often than men (58.6%; 320/546), but that older children were more likely carried by men. Though back loads are less metabolically costly than side loads, they observed most infants being carried on the side (*Abe, Yanagawa & Niihata, 2004*; *Rhinegold & Keen, 1963*). Side carrying allows accessible interaction between child and parent (*Sallstrom, Snyder & Wall-Scheffler, 2012*).

Humans' speed decisions when walking with others and carrying children provides insight into the adaptive influence of social relationships. Understanding the circumstances under which people deviate from their optimum speeds reveals the energy tradeoffs and social interactions in which people engage. However, the dynamics of how walkers adjust their speed in varied groups and in groups containing children remains unexplored. Furthermore, little ecological data exists to illustrate the relationships between walking speed and child-carrying. Because the determinants of group dynamics and parental investment are partially cultural, the present study examines walking behavior between samples from the Northwestern United States and Central Uganda. Traditional Ugandan culture tends toward communalism and high-contact, whereas traditional United States culture is individualistic and low-contact. We hypothesize that, across cultures, walking with or carrying children will result in significant decreases in walking speed compared to walking with adults or carrying comparable loads of food or goods.

## METHODS

### Subjects

A total of 1,721 subjects were observed in metropolitan public areas walking alongside roads. 969 subjects (355 male and 614 female) were observed in Central Uganda and 752 subjects (337 male and 415 female) were observed in Washington State, US. All subjects observed purposefully walking in a steady state toward a destination (i.e., not for exercise) and for whom we could monitor their entire trajectory between the two markers were

**Table 1   Focal subject age categorization guidelines.** Average height measurements calculated as the average height of American males and females in that designated age group according to the 2016 CDC Anthropometric Reference (*Fryar et al., 2016*).

| | |
|---|---|
| Child | Estimated age 3–12 years; immature facial features, short stature, and commonly accompanied by older persons |
| Teenager | Estimated age 13–18 years; average height 1.67 meters; older school-aged individuals, often carrying backpacks |
| Adult | Estimated age 19–59 years |
| Older adult | Estimated age 60+; characterized by greying hair, developed wrinkles, and/or stooped posture |

included. Persons walking at a brisk pace and wearing fitness attire without additional medium to large loads were judged to be walking for exercise. Only subjects who walked uninterrupted through the observation area and of whom the observer had an unobstructed view were included. This sample included persons walking alone or in groups, as well as those walking unloaded or loaded. Subjects reflected a normal range of body sizes, which has been shown to be similar for American and Ugandan samples (*Burgess & Burgess, 1964*; *Fryar et al., 2016*). All procedures were approved by Seattle Pacific University's IRB Committee; IRB #151606011.

## Procedure

From a removed viewing position, an observer recorded the subject's speed as the time taken to walk between pre-measured stationary markers. We used a stopwatch to measure time between the markers, as this is considered the gold-standard method of monitoring pedestrians (*Wagnild & Wall-Scheffler, 2013*; *Amato, 1983*; *Arango & Montufar, 2008*; *Bohannon, 1997*; *Bornstein & Bornstein, 1976*; *Coffin & Morrall, 1995*; *Elman, Schulte & Bukoff, 1977*; *Gates et al., 2006*; *Knoblauch, Peitrucha & Nitzburg, 1996*; *Korte & Grant, 1980*; *Koushki, 1988*; *Tarawneh, 2001*). The stationary markers measured 7.3 m apart in the Ugandan location and 9.7 m apart in the United States location. These distances, as well as the position of the observer, were consistent across all collection days. The observer started the stopwatch when the subject's estimated center of mass crossed the first measured mark, and stopped the stopwatch when it passed the second measured mark. Only people who walked close to the markers were measured, and thus potential problems due to parallax were minimized. Speed in meters per second (m/s) was determined by dividing the marked distance by subject's crossing time.

The observer recorded the sex and general age category of the subject (i.e., child, teenager, adult, or older adult; Table 1). Any loads carried or pushed by the subject were categorized by **type** (i.e., child, food, goods, or stroller), **position** (i.e., front, back, side, or shoulder), and **size** (i.e., small, medium, or large) of loads carried by the subject. Strollers, while not carried on the arms or torso like other loads, have been shown to significantly moderate speed and were of interest as an infant-transport device (*Alcantara & Wall-Scheffler, 2017*). A small load was a purse or a small bag in the hands; a medium load was a torso-sized bag such as a backpack; a large load was an oversized bag (Table 2).

**Table 2  Load categorization guidelines with example load descriptions and masses for loads of each size and type category.** Child mass measurements calculated as the average mass of American males and females in that designated age group according to the 2016 CDC Anthropometric Reference (*Fryar et al., 2016*).

| Load size | Type | Description | Approximate mass |
|---|---|---|---|
| Small | Food | Coffee cup or beverage; individual snack food items able to be handled with one hand. | <1 kg |
| | Goods | Thin books, paper folders, clutch-sized purse | <1 kg |
| Medium | Food | A single, full bag of groceries, bundle of food for sale approximately of torso size; visibly full | 1–6 kg |
| | Goods | Bag or mid-sized purse approximately of torso size | 1–6 kg |
| | Child | Infant (unable to mobilize independently) | $\bar{x} \approx 6.5$ kg |
| Large | Food | Packages of groceries or food for sale larger than torso-sized | >6 kg |
| | Goods | Oversized bag (larger than torso size) | >6 kg |
| | Child | Toddler or Child (able to mobilize independently) | 11.5–19.7 kg |
| | Stroller | Child-carrying device pushed in front of walker | 13 kg (stroller alone) plus 6–15 kg (size of child) |

When the load carried was a human child, the relative age category (i.e., infant (<1 year old), toddler (1-3 years old), or child (>3 years old)) of the child was recorded. These related to the size of other loads with infants being classified as a medium load, and toddler and children being classified as large loads (Table 2). Finally, the observer recorded the gender and age category of all other individuals accompanying the focal walker. When adults were observed walking in groups, the observer chose the group member closest to the markers as the focal figure for data collection (see *Zivotofsky & Hausdorff, 2007* for synchronous walking speeds between partners). Children were selected as the focal figure only when walking alone or in groups of only children.

All Ugandan and American subjects were observed at a single viewing position within the respective country. In Uganda, that location was 0°21′38.2″N, 32°44′48.3″E. This was a walking path along Kampala-Jinja Road in Mukono, Uganda. In the United States, the location was 47°38′18.3″N 122°21′22.9″W. This was also a walking path (pedestrian sidewalk), located in Seattle, Washington.

In both locations, (i.e., the Ugandan and American streets) subjects were observed walking in central "errand running" areas near grocery stores and shopping centers over flat ground. Neither location crossed a crosswalk. Both areas had wide, beside-roadway paths such that subjects could comfortably walk abreast in groups. In the United States, subjects walked on paved concrete, while Ugandan subjects walked on packed, smooth dirt. These conditions were typical of the environment; we chose to match circumstances as closely as possible (that is, style and purpose of walking) which meant we were not able to match substrate perfectly. All data were collected in the early or middle afternoon. Weather conditions during each observation period were also recorded, though did not have any significant influence on any model.

## Data analysis

Data were analyzed using SPSS Statistics via a univariate general linear model with speed as the dependent variable. The average temperature during collections times was 21 °C

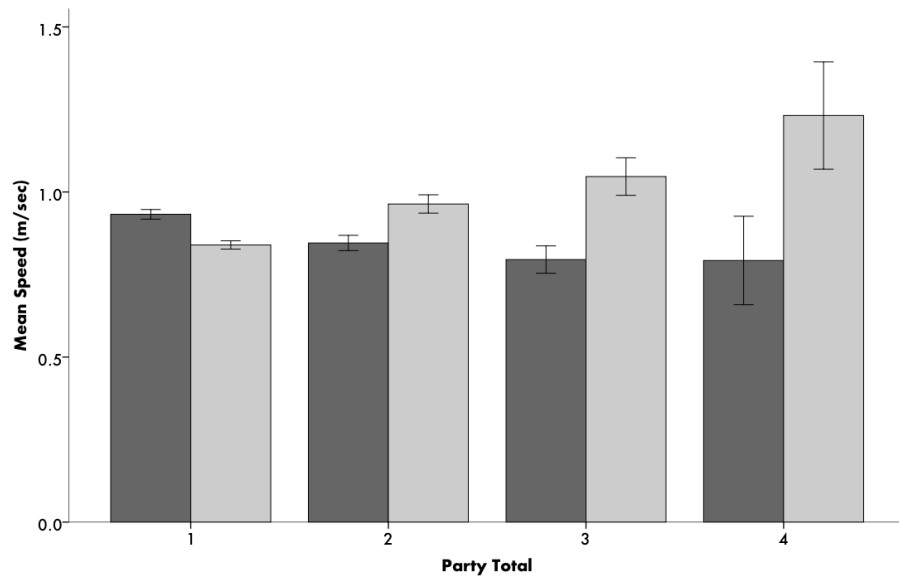

**Figure 1** Average speeds of Ugandans (dark grey) and Americans (light grey) based on walking group size (*x*-axis refers to number of people walking in the group). Error bars indicate 95% CI.

in Uganda and 14° Celsius in the United States. Weather conditions did not significantly impact walking speeds ($p > 0.197$). Data were split by location (i.e., Ugandan observation site or United States observation site) when analyzing effects within a sample. Walker-gender, type of load carried, group composition, number of group members, and the presence of children in the group were tested as factors against the dependent variable of walking speed. Group composition was coded according to the following categories: alone, only men in the group (including walker whose speed was being monitored); only women in the group (including walker whose speed was being monitored); only adult men and women; men and children; women and children; men, women, and children; and only children. Because of the small sample of older adult subjects, they were included within the "adults" category. Any category including children accounted for groups in which children walked freely or were carried.

Tukey post-hoc tests were done on each factor.

## RESULTS

The effects of the type of load carried, the presence of children, and group composition on walking speed each differ significantly between people walking in Central Uganda and people walking in the West Coast of the United States ($p < 0.001$). Across all walking groups, Ugandans on average walked 2.6% faster than Americans ($p < 0.01$). However, while Ugandans walking alone walked 11.3% faster than Americans walking alone (Table S1), Ugandan walkers in groups were slower than both American groups (18.3% slower) and individual Ugandans (11.7% slower) (Fig. 1, $p < 0.001$). In Uganda, speed decreased with group size, while speed increased with group size in America (Fig. 1, $p < 0.001$ for

**Table 3  Count and mean, minimum, and maximum speeds of walkers in Central Uganda and West Coast United States.** No subjects walked with strollers in the Central Ugandan population.

| | | Central Uganda | | | | W.C. United States | | | |
|---|---|---|---|---|---|---|---|---|---|
| | | Count | Mean speed (m/s) | Min. | Max. | Count | Mean speed (m/s) | Min. | Max. |
| **Unloaded adults alone** | Male | 40 | 1.04 | 0.61 | 1.59 | 108 | 0.8 | 0.58 | 1.23 |
| | Female | 14 | 0.89 | 0.71 | 1.22 | 37 | 0.84 | 0.56 | 1.23 |
| **Load condition** | Unloaded Male | 48 | 1.02 | 1.59 | 0.53 | 143 | 0.84 | 1.44 | 0.57 |
| | Unloaded Female | 16 | 0.90 | 1.22 | 0.71 | 64 | 0.90 | 1.29 | 0.56 |
| | Loaded Male | 307 | 1.00 | 1.69 | 0.45 | 194 | 0.87 | 1.66 | 0.53 |
| | Loaded Female | 598 | 0.84 | 1.44 | 0.38 | 351 | 0.89 | 1.66 | 0.54 |
| **Load type** | Child Male | 11 | 0.81 | 1.00 | 0.62 | 4 | 1.06 | 1.28 | 0.96 |
| | Child Female | 132 | 0.77 | 1.08 | 0.38 | 11 | 1.06 | 1.21 | 0.89 |
| | Food Male | 69 | 1.01 | 1.36 | 0.62 | 15 | 0.85 | 1.12 | 0.71 |
| | Food Female | 90 | 0.88 | 1.44 | 0.47 | 37 | 0.90 | 1.45 | 0.66 |
| | Goods Male | 227 | 1.00 | 1.69 | 0.45 | 171 | 0.87 | 1.66 | 0.53 |
| | Goods Female | 376 | 0.86 | 1.38 | 0.46 | 287 | 0.88 | 1.66 | 0.54 |
| | Stroller Male | 0 | | | | 4 | 0.99 | 1.14 | 0.81 |
| | Stroller Female | 0 | | | | 16 | 0.95 | 1.16 | 0.77 |
| **Subject party** | Alone Male | 258 | 1.03 | 1.69 | 0.53 | 273 | 0.83 | 1.56 | 0.53 |
| | Alone Female | 405 | 0.87 | 1.38 | 0.38 | 278 | 0.84 | 1.25 | 0.54 |
| | Group Male | 97 | 0.91 | 1.54 | 0.45 | 65 | 0.96 | 1.66 | 0.63 |
| | Group Female | 209 | 0.80 | 1.44 | 0.43 | 136 | 1.00 | 1.66 | 0.61 |
| **Children present** | No children Male | 329 | 1.01 | 1.69 | 0.53 | 303 | 0.85 | 1.66 | 0.53 |
| | No children Female | 432 | 0.87 | 1.44 | 0.46 | 351 | 0.87 | 1.66 | 0.54 |
| | Children present Male | 26 | 0.82 | 1.54 | 0.45 | 35 | 0.98 | 1.33 | 0.66 |
| | Children present Female | 182 | 0.78 | 1.21 | 0.38 | 63 | 1.01 | 1.39 | 0.74 |

both). Sixty-eight percent (663/969) of Ugandans walked alone, as did 73% (551/752) of Americans (Table 3). Men walked 18% faster than women in Uganda ($p < 0.01$), while American men and women walked at about the same speeds ($p = 0.411$).

## Load

In both locations, women were much more likely to be loaded than were men (Table 3; $p < 0.001$). Ugandans walked 10% more slowly when loaded, while Americans walked 3.4% faster when loaded ($p < 0.001$ for both). This trend held true in both locations for loads of all sizes. Goods (e.g., purses, backpacks, or items for sale) were the most common load type for both locations, constituting 73% ($n = 1,061/1,450$) of loads carried. In both locations, food and goods, but not children, were carried at similar speeds when controlling for size (Fig. 2; $p < 0.01$). Central Ugandans walked significantly faster when back-loaded than when front loaded ($p < 0.05$). In America, people walked slower when back-loaded (e.g., with a backpack) than when front-loaded ($p < 0.001$).

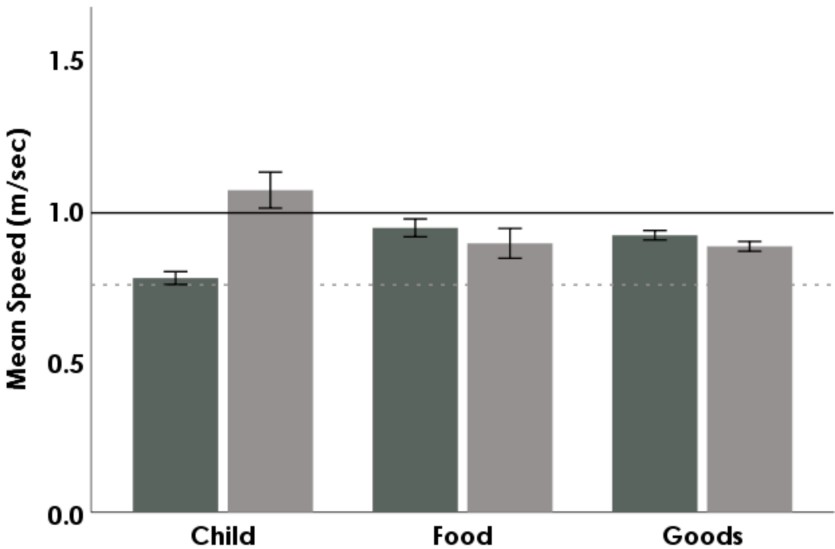

**Figure 2** **Average speeds of Ugandans (dark grey) and Americans (light grey) carrying either children, food, or goods.** Error bars indicate 95% CI. The solid black line represents the unloaded mean for Ugandans; the dotted grey line represents the unloaded mean for Americans.

## Children

Of the 1,449 people carrying loads, 158 carried children (11%). The effect of carrying children on walking speed was dependent upon location ($p < 0.001$). Ugandans walked between 15.4–17.6% more slowly when carrying children than when carrying food or goods (Fig. 2; $p < 0.01$), while Americans walked between 19.8–21.2% more quickly (Fig. 2; $p < 0.01$). These relationships remained significant when child loads were compared to food or good loads of the same size category ($p < 0.01$). The Americans' speed did not differ significantly amongst food, goods or strollers ($p > 0.05$). Women (143/158) were overall more likely to carry children than were men (15/158; $p < 0.01$). American children were more often carried on the front ($n = 11/15$) than the side ($n = 3/15$) or back ($n = 1/15$). Ugandan children were most frequently carried on the front ($n = 57/143$) or the side ($n = 56/143$) of the subject. While Ugandan women carried children on their fronts, backs, or sides, Ugandan men never carried children on their backs but only on the front or side. Position of the child load was not significantly correlated to speed in either location ($p > 0.05$).

Amongst Ugandans, women carried infants ($n = 100/133$) more often than toddlers ($n = 31/133$) and children ($n = 2/133$), while men carried toddlers ($n = 7/11$) more often than infants ($n = 4/11$). American women and men did not carry infants at a higher rate than toddlers. Age category of the child-load had no significant relationship to speed ($p > 0.05$).

Whether children were present in the walking group significantly influenced speed ($p < 0.001$); however, once children were present ($n = 304$), there were no significant differences between whether children were carried, walked by themselves, or were pushed

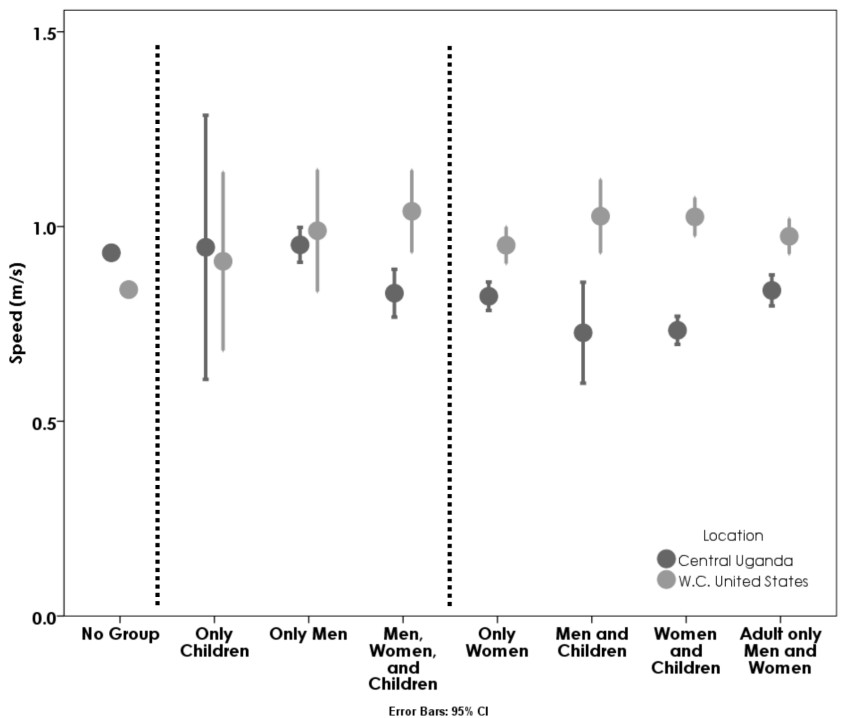

**Figure 3 Average speeds of Ugandans (dark grey) and Americans (light grey) based on walking group composition.** Both group composition and location had a significant impact on speed ($p < 0.001$; $p = 0.036$), and significantly interacted ($p < 0.001$). The dashed lines demarcate Tukey's post-hoc results from our GLM for Ugandan walkers—the speeds in the bracket adjacent to the single walkers were not significantly different from them. For American walkers, only children walking together had walking speeds similar to adults walking alone (American adults walked more slowly when walking alone; children together were similar to this). Error bars indicate 95% CI. The confidence intervals for the walkers without a group are so small they are covered by the dot itself.

in a stroller ($p = 0.168$). American groups walked faster when children were present, while Ugandan groups walked more slowly when children were present ($p < 0.001$).

## Group composition

The effect of group composition on walking speed was highly dependent on the sample location (Fig. 3; $p < 0.001$). Amongst Ugandans, the walking speeds of groups with men, women and children, men-only groups, and groups of children are not significantly different from the walking speeds of individuals alone ($p > 0.267$). All other group compositions (see Fig. 3) walk more slowly than people walking alone ($p < 0.05$), but do not show any significant differences from each other (e.g., groups of men with children walk about the same pace as groups of women with children—which is quite slow). Amongst Americans, groups of only children walked at similar speeds to individuals walking alone ($p > 0.986$) while all other group types walked faster than individuals walking alone ($p < 0.01$). There were no significant differences in speed between any American group types.

### Sex differences in group walking

While group composition was significantly correlated to speed in both samples ($p < 0.05$), there was a significant interaction between walker sex and their group composition only in the Ugandan sample ($p < 0.01$). In Central Uganda, men alone walk significantly faster than men walking with other men ($p < 0.05$) while men alone in the West Coast United States walked significantly slower than men walking with other men ($p < 0.01$). Ugandan women tended to walk more slowly in groups of women than when alone ($p = 0.10$), but American women walked faster in groups of women than when alone ($p < 0.001$). Each of these analyses included lone male or female subjects from all age groups.

## DISCUSSION

In the present study, we observe that Ugandan and American walkers choose different speed strategies based on universal situations—that is, just because child carrying and group walking are human universals, it is clear that there are not universal solutions to these situations! While American walkers tend to increase their speeds when walking in groups, Ugandan walkers decrease their speeds when walking with others. Ugandans alone walked faster than Americans alone, even though all group types of Ugandans walked slower than their American counterparts. These differences in speed decisions also extended to individuals walking alongside or carrying children; Ugandans slow down when children are present, while Americans speed up. In both samples, children were carried at significantly different speeds than similarly-sized loads of food or goods.

It is worth noting that our adult walking speeds are slower than some speeds previously recorded in the western contexts (see *Ishaque & Noland, 2008* for review of speed data from across 10 studies). Substantial ethnographic data, however, show speeds similar to those we found here, particularly among people walking in groups carrying loads typical of those seen here. Female !Kung (*Bentley, 1985*) and Xhosa (*Lloyd et al., 2010*) foragers, for instance, both walked around 0.9 m/s, Hadza women walked at 0.97 m/s (*Marlowe, 2006*), and Ache women walked at 0.8 m/s (*Hurtado et al., 1985*). We believe that our data are more comparable to the foraging data, rather than to other published urban data due to our location, which is oriented around a common goal or 'function' of the walking behavior; that is, in our sample, people were walking towards or away from a market/gathering area which is similar to the goal and location of typical speed snapshots of foragers. We did not collect data across a cross walk, nor in a central downtown area in which people are highly compacted and moving to and from work and/or lunch breaks. These are the locations typical of many speed-related studies on urban populations and most likely represent more rapid walking. In other work we have also collected data on people crossing streets, and these data show people walking an average speed of 1.4 m/s, though loaded people walk more slowly than this (*Bonner-Harris et al., 2018*). Further studies comparing and contrasting walking speed in the same city at different intersections will be an obvious next step for this research.

Our data also show that American walkers in groups walk faster than those walking alone. Though pedestrians generally slowdown in groups (*Wagnild & Wall-Scheffler, 2013*;

*Costa, 2010*; *Frimenko, Goodyear & Bruening, 2016*; *Moussaid et al., 2010*), some data do indicate that individual and paired walkers walk at equivalent speeds, though this is clearly situationally dependent (*Tarawneh, 2001*). Our findings highlight the specific nature of our "errand-running" observational setting; larger groups may additionally be associated with higher task demand and hurriedness in this naturalistic setting.

Selecting a speed for a particular walking task involves a complex set of interactive variables that include both physiological (metabolic energy expended, heat load, and water loss) as well as behavioral (time spent on the task and the possibility of socialization) variables (*Wall-Scheffler & Myers, 2013*; *Wagnild & Wall-Scheffler, 2013*; *Wall-Scheffler & Myers, 2017*). Because of the habitual nature of walking, small differences in these costs will accumulate into the large changes that over time that influence fitness (*Gibson & Mace, 2006*). Speed adjustment decisions, then, are expected to be the product of selective tradeoffs for minimizing costs and thermoregulatory burdens, while maximizing task accomplishments (*Wall-Scheffler & Myers, 2013*; *Miller et al., 2012*). While previous studies by Bornstein and Bornstein (*Bornstein & Bornstein, 1976*) and Levine and Norenzayan (*Levine & Norenzayan, 1999*) compare walking speeds across 6 and 31 countries, respectively in a variety of contexts, we provide higher resolution data demonstrating cultural variation in the driving forces behind walking behaviors.

Ugandans in groups, for example, accept a higher time cost by selecting slower speeds in groups than alone. This strategy has been widely observed in ecological (*Costa, 2010*; *Moussaid et al., 2010*) and controlled walking studies (*Wagnild & Wall-Scheffler, 2013*; *Frimenko, Goodyear & Bruening, 2016*). Two main reasons have been suggested as to why people choose to slow down when walking together. First, differences in optimum walking speed based on size and sexual dimorphism generally lead to faster group members deviating from their energetically optimal speed to accommodate slower walkers in the group (*Costa, 2010*). Second, slower walking speeds are correlated to closer interpersonal distances between walkers, such that any increased costs from walking more slowly may be outweighed by the benefit of social investment and bonding (*Wagnild & Wall-Scheffler, 2013*; *Costa, 2010*; *Wellens & Goldberg, 1978*). Faster walkers are less aligned with their walking partners (i.e., they are more staggered), so interpersonal contact may have to be sacrificed to walk quickly. Our data for Ugandan mixed sex groups, groups of women, and groups with children are consistent with these two explanations. Our finding, though, that Ugandan men walk more slowly with other men than when alone challenges an existing framework for men. It has been accepted that many Western men speed up when walking with other men due to a high societal emphasis on hierarchy and competition among males (*Wagnild & Wall-Scheffler, 2013*; *Costa, 2010*; *Boles, 1981*). A difference in male-male relationship dynamics would explain the slower walking speeds, and potentially psychological closeness of Ugandan men.

Gendered relationship norms also clearly vary across cultures. These norms prescribe appropriate interactions between genders, including interpersonal distances. *Baxter (1970)*, for instance, finds that interpersonal distances of same- and mixed-sex pairs differed based on ethnicity in the United States. Traditionally gendered traits, such as competitiveness, have also been shown to vary culturally. Amongst the matrilineal Khasi, for example,

women show higher competitiveness than men, while men tend toward competitiveness in patrilineal cultures (*Gneezy, Leonard & List, 2009*). The Baganda, Central Uganda's predominant ethnic group, are a patrilineal group like most Western cultures (*Wyrod, 2008*); however they are a high-contact culture (*Awa, 1988*; *Watson, 1970*). High-contact cultures, like the Baganda, have been shown to interact at closer distances than people from low-contact cultures like the United States (*Remland, Jones & Brinkman, 1995*). Uganda's numerous additional ethnic groups similarly follow high-contact tendencies. While residents of the Northwestern United States originate from diverse ethnic groups of both low- and high-contact natures, American cultural norms align with low-contact values. Tendency toward closer interpersonal contact may explain why all Ugandan walking groups, including groups of men, slow down when walking together–because slowing down leads to closer contact between walkers (*Costa, 2010*).

We also find that, in both our Ugandan and United States samples, children are carried at significantly different speeds than other load types. That children are transported differently than other loads is widely recognized (*Kramer, 2004*; *Kramer, 1998*). *Hodges & Lindhiem (2006)* show that walkers carrying children are perceived as more cautious than those carrying groceries, regardless of their actual gait. Infant carrying emerged early in the primate lineage, whereas foraging-related burden carrying has been thought to emerge in early members of the genus *Homo* (*Ross, 2001*; *Rhinegold & Keen, 1963*; *Leonard & Robertson, 1997*). We can expect then, that these two types of load-carrying evolved under different constraints. Our data show that children are carried at faster speeds than other loads in the North American sample, but at slower speeds in the Ugandan sample. *Kramer (1998)* predicts that mothers should carry their infants–rather than allow them to walk independently–when the carrying mother's energy expenditure is less than that of the independently walking pair. At faster speeds, she argues, it is beneficial to carry the child. Our data, however, show that groups with individuals carrying children walk at similar speeds to groups with children independently walking. Additionally, Ugandans also carry children at slower speeds (and not at faster) which seems to conflict with Kramer's model (*Kramer, 1998*). Either Ugandan's choices are influenced by parental care norms not addressed in Kramer's model, or the nature of Ugandans' walking task differs from Americans' and those predicted by Kramer's model. As discussed above, interpersonal contact is more normative in Baganda culture than in Western cultures. Infants in African groups such as the !Kung or Gusii are held or touched about 70–80% of daylight hours, compared to 12–20% in industrialized nations (*Hewlett, 1996*). Higher contact norms unaccounted for in Kramer's efficiency model (*Kramer, 1998*) may impact the decision to carry a child, even at slower speeds. It is also possible that Ugandans choose to carry at slower speeds because their errand-running task requires walking longer distances. In this case, it is energetically favorable to incur child-carrying costs because allowing a child to walk over long distances will accumulate into greater total energy expended for the pair in addition to the larger time-costs considering the lower optimal walking speeds of children (*Cavagna, Franzetti & Fuchimoto, 1983*; *DeJaeger, Willems & Heglund, 2001*).

## CONCLUSION

The differences in the walking speeds of groups and when carrying children between Central Ugandan and Northwest United States samples accentuates the role of culture and environment in mediating decisions that have energetic consequences. In the future, greater efforts should be taken to understand the ways in which walking behaviors vary across cultures (see *Bornstein, 2002*). Our data also show that existing predictions of how humans will respond to locomotor or mobility challenges are specific to one sample and may not generalize across cultural groups. It is also important to recognize the ways in which behavioral differences between groups, either between sexes or across populations, are influenced by external social factors such as dominance structures in addition to innate biological or morphological differences (e.g., *Travis & Yeager, 1991*; see *Dingwall et al., 2013*). Such recognition builds an understanding that behaviors are environmentally situated processes rather than static attributes.

## ACKNOWLEDGEMENTS

We offer many thanks to A Belleville, BR Apouli, and BA Asiimwe for their assistance in data collection and to A Luthi for her help with data entry. K Neuhouser and MJ Myers gave helpful comments that have greatly improved the paper.

### Funding

The authors received no funding for this work.

### Competing Interests

The authors declare there are no competing interests.

### Author Contributions

- Leah Bouterse conceived and designed the experiments, performed the experiments, contributed reagents/materials/analysis tools, authored or reviewed drafts of the paper, approved the final draft.
- Cara Wall-Scheffler conceived and designed the experiments, analyzed the data, contributed reagents/materials/analysis tools, prepared figures and/or tables, authored or reviewed drafts of the paper, approved the final draft.

### Human Ethics

The following information was supplied relating to ethical approvals (i.e., approving body and any reference numbers):

All procedures were approved by Seattle Pacific University's IRB Committee; IRB # 151606011.

### Data Availability

The raw data are provided in a Supplemental File.

## Supplemental Information

Supplemental information for this article can be found online at http://dx.doi.org/10.7717/peerj.5547#supplemental-information.

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
