# Peer review of "Children are not like other loads: a cross-cultural perspective on the influence of burdens and companionship on human walking"

_PeerJ, doi:10.7717/peerj.5547_

## Round 0.1 · original submission · Minor Revisions

I apologize for the longer than normal period of review time we have taken on this manuscript. There is some variation in reviewer recommendations but having read through the comments and the manuscript, I feel that minor revisions are required at this stage. In particular, please do carefully address in your rebuttal (and, where appropriate, edit the manuscript) the concerns of Reviewer #1; I feel that these changes may be a matter of language and clarification rather than major manuscript revisions. Also, please address the minor issues raised by Reviewer #2.

·

Basic reporting

The manuscript is very well written. It uses scientific language of the right level and the manuscript will be understandable for anyone with a basic science background who is interested in it.
The Introduction is excellent and provides a very good overview of the background, i.e. what determines gait and specifically walking speed across environments, genders, social contexts etc. I feel this is fascinating to read as it is very relevant not just to biomechanists but also to e.g. cultural and physical anthropologists, and psychologists.
The figures and tables are suitable. I checked the raw data and it is clearly presented ad an Excel sheet.
The Results and Discussion refer nicely to the Introduction. The paper in not too long nor too brief for the contents.
What I really appreciate about this paper is that it uses the simplest techniques possible – anyone with a timer and a roll of marking tape could do this! The manuscript shows that there is still scope for highly relevant research without having to rely on high-tech. What suffices, in this case, is to have an innovative and scientifically interesting idea, which the authors have. This is a refreshing manuscript that is a pleasure to read.

Experimental design

There is primarily an observational study, but the data are well used to test specific hypotheses set out in the Introduction. The main purpose of the study is to compare how various loads are carried across cultures and the authors have highlighted the importance of such research very clearly and describe how their data fill a knowledge gap. The data have been collected to a high technical and documented ethical standard. The research could be replicated with the information provided in the paper, although the description of the study sites (lines 139-141) should be more detailed (names and concrete details of locations, GPS coordinates).
I do have some concerns about the actual data which I hope can be clarified by the authors in a revised version. Whilst the number of subjects is high (1721), the authors used a single site per “culture” (North American vs. Ugandan). Whilst they justify their choice, it can be asked whether a single suite can correctly represent the culture. This concern is amplified by the absolute speeds of the participants being much lower than described in virtually any previous study. There are some other uncommon findings such as children walking at the same speed as adults (see section 3 below). Therefore, at this stage, I am not convinced that the speeds recorded are representative of the populations measured. Whilst the paper focuses on _relative_ speed differences between conditions and populations, I feel that if the _absolute_ speeds (even for individuals walking alone) deviate from normal, the relative differences might be skewed as well.

Validity of the findings

The findings are interesting for scientists from different backgrounds and the analysis as such (statistics etc.) is valid. As outlined previously, it should be justified better why the raw data are representative of “American“ and “Ugandan” cultures even though only one site per culture was used, and some aspects of the data suggest they are not what would normally be expected based on a fairly large body of existing literature.
The clearest example is the self-selected walking speed of the American adults in this paper. Whilst the text discussed only relative numbers, the absolute speeds can be seen in the figures and in Table 3. They are nearly always below 1 m/s. This is very slow for self-selected walking speed and well below what would be the optimum walking speed in terms of Cost of Travel. The mean speed for the American adults is 0.8 m/s. The authors very briefly mention this in the Discussion (lines 283-288) and suggest the speeds are similar to some ethnographic data (female foragers etc.) but I would argue that neither these populations, nor their walking contexts, are good bases of comparison for the subjects in the current study. Levine & Norenzayan (1999) and Ishaque & Noland (2008), both cited in the manuscript, report on comparable populations to the Americans and they find self-selected walking speeds of typically around 1.4 m/s. This observation is my main concern about the manuscript and I feel this issue should be tackled convincingly by the authors as it might affect the results and their significance.
A smaller but not insignificant concern is that the substrate is different between the sites; concrete in the US and packed dirt in Uganda. It is a pity that the same substrate (concrete) was not used in both locations and some rationale should be given why this is judged to have no relevant effect on the data (if so).

Additional comments

Here I offer some additional and more minor comments to the authors which might help them to further improve the manuscript.

- An issue that I hope can be clarified is seen in the raw data but is not discussed in the manuscript itself. When checking the raw data, apart from the first author there is an additional observer “A. Bellville” (who is currently mentioned in the Acknowledgements) and potentially another one labelled “3”. The Acknowledgements mention more people involved in data collection (but not mentioned in the raw data). Have the authors checked for a potential observer bias? Were these observers crucial to the study (have they had intellectual input in it?) and if so do some of them deserve co-authorship? I have no reason at all to doubt the academic judgment on authorship of the authors, but I suggest that at least observer bias is tested. The authors have argued that the American and Ugandan populations are similar in terms of body size and that the contexts are also comparable. Yet, they might want to provide additional arguments why the difference between the populations is just (or mostly) cultural. E.g. did the American subjects include African Americans? Are they different? Please provide some references for the statements about culture in lines 130-133.
- Line 106: “closer to the CoM” – but carrying on the head in African women and Nepalese porters is very efficient?
- Line 158: “estimated” center of mass.

Reviewer 2 ·

Basic reporting

The language is clear and professional throughout. I saw no typographical or grammatical errors. The article is also structured in a straightforward way. Data is shared effectively by means of graphs, tables and the main text, and discussion is focused on the data itself.

Experimental design

The primary research on walking speeds, group size and particularly child-carrying amongst American and Ugandan walkers is original and significant. The research questions derived from this are all well defined, and a range of relevant literature is referred to in the opening part of the paper and the discussion. Methods are described clearly and there are no problems with technical or ethical standards.

Validity of the findings

The findings are clearly stated in the results section and their meaning and implications are discussed in the following section. Of particular interest was the finding that child caring groups in Uganda walk slower, as do men walking in groups, compared to in America. This is explained by way of the possibility of a higher person-to-person contact norm in Uganda, and the relative lack of male competitiveness in America. These are both intriguing argument that support the general hypothesis of the paper that culture is a significant factor in walking behaviour. The conclusion to the article is clearly put in this regard to suggest alternatives to biological and morphological difference as explanatory models.

Additional comments

In general I found this to be a very readable and interesting paper. It would be interesting to compare these observational results with some qualitative data exploring the walkers' own perceptions of why they walk in the way they do.

Reviewer 3 ·

Basic reporting

Review: Bouterse, L. and Wall-Scheffler, C.; Children are not like other loads: A cross-cultural perspective on the influence of burdens and companionship on human walking.
Submission to: PeerJ.
This manuscript is an analysis of variation human walking behaviors, including load carrying and child transport, derived from non-invasive observations made in two urban public settings: Central Uganda; the United States. The researchers have recorded walking speed, load carriage, and group composition and specifically highlight distinctions in group walking and load carrying behaviors seen between the two cultural settings. suggest that the different location, thereby concluding that cultural factors play a role in altering the dynamics of human locomotion in different world settings.
The current manuscript is acceptable for publication in PeerJ as it addresses several unexplored topics of interest to locomotor researchers, namely what impact may the carrying of children as well as group composition have human locomotor behavior and are the dynamics of walking and load carrying dependent on cultural context. The authors demonstrate that the answers to such questions are yes, locomotor dynamics are influenced by cultural factors and that future models of human walking behaviors should attempt to account for cultural context. The current manuscript is a tightly streamlined paper. It is succinctly written with an introduction that clearly states the goals of the study, a methodology that provides the necessary descriptions for the variables of interest and those used, a results section highlighting the clear differences and similarities between the two groups sampled and analyzed, and a discussion section that contextualizes the implications of these results without overstating them. The implications of the current study should have an influence on future locomotor research by indicating that cultural context is an important factor rather than an unquestioned practice used in most current locomotor models that “one size fits all.”. While the study could have been enhanced with subsequent controlled locomotor observations and analyses of the individuals initially examined, the non-invasive methodology created limits on such follow-up. Regardless, the current paper does in fact stand on its own with respects to its results and conlusions. Thus, it should be published.

Experimental design

For this particular observational study, the research design holds.

Validity of the findings

The authors have not overstated the implications of their results.

Additional comments

See basic reporting.

---

## Round 0.2 · accepted · Accept

Thanks for addressing the concerns of reviewers and adding some additional text (and changing populations to samples) to exhibit consideration of issues raised by reviewers that might be shared with readers.

#